# Evaluation of IL-33R and Galectin-3 as New Biomarkers of Cardiac Damage after Polytrauma—Association with Cardiac Comorbidities and Risk Factors

**DOI:** 10.3390/jcm11216350

**Published:** 2022-10-27

**Authors:** Birte Weber, Maika Voth, Katrin Rottluff, Ingo Marzi, Dirk Henrich, Liudmila Leppik

**Affiliations:** Department of Trauma-, Hand- and Reconstructive Surgery, University Hospital Frankfurt, Goethe-University, 60596 Frankfurt am Main, Germany

**Keywords:** cardiac damage, polytrauma, biomarker, galectin-3, IL-33 receptor, cardiac risk factors, comorbidities

## Abstract

Polytrauma is one of the disorders with the greatest economic impact on healthcare in society and one predictor for poor outcome is cardiac damage. Interleukin 33 receptors (IL-33R) and galectin-3 are two new potential cardiac trauma biomarkers that are the subjects of this investigation. Additionally, this study assesses pre-existing cardiac damage or risk factors as predictors of cardiac damage after polytrauma. This retrospective study includes 107 polytraumatized patients with an ISS ≥16 admitted in a Level 1 Trauma Centre. Plasma samples were taken at admission. IL-33R and galectin-3 concentrations were detected in plasma samples by ELISA. Both did not correlate with the cardiac damage measured by troponin. Next to troponin, IL-33R was increased in patients with pre-existing cardiac comorbidities. In the subgroup of patients with cardiac comorbidities, the BMI and the initial blood sugar level were significantly increased compared to patients without cardiac comorbidities. Galectin-3 and IL-33R were shown to not correlate with cardiac damage. However, our data suggests that IL-33R protein should be revised in future studies as a marker of cardiac comorbidities. Further, our data indicate that patients with cardiac comorbidities represent a separate group of polytrauma patients characterized by higher concentrations of troponin, IL-33R, BMI and initial sugar level.

## 1. Introduction

Severe trauma remains one of the disorders with the greatest healthcare and economic impact in society. Particularly in polytraumatized patients, a quick, precise and correct recognition of presenting injury patterns is important for proper patient management, as a delayed diagnosis may cause secondary complications and exaggerate mortality [1,2]. In addition to lung injury, cardiac damage was identified as a predictor of a poor outcome after trauma [3]. In general, cardiac injuries are associated with dysrhythmias, ventricular fibrillation, impaired cardiac function and sudden cardiac arrest [4]. Heart injuries following multiple traumas are characterized by a prolonged time on ventilation, as well as by a longer hospital stay [5]. Myocardial damage presents a wide range of symptoms, from asymptomatic to death, based on the severity and mechanism of the injury. Due to the variability of symptoms, the diagnostics of cardiac contusion are complicated and often based on laboratory markers, such as systemic troponin T [6]. Despite the widespread use of troponin T as a marker, the mechanisms of cardiac troponin elevation are extremely numerous and not limited to myocardial damage [7]; this stimulates further research into reliable early markers of traumatic cardiac contusion. Therefore, the present study investigates two new potential biomarkers in the diagnosis of cardiac trauma: IL-33R and galectin-3. Both are discussed in the literature as new biomarkers of early cardiac damage, in acute heart failure and acute coronary syndrome [8,9].

Next to the evaluation of the cardiac biomarkers, this study hypothesizes that pre-existing cardiac damage or risk factors could be potential predictors of cardiac damage after polytrauma. The most frequent cause of death in elderly comorbid individuals surviving more than 48 h after trauma is sepsis or coronary artery disease. Wilson, et al. highlighted in a 1994 study that 30% of trauma patients over 55 years old have pre-existing diseases, and by the time these patients reach 65 years of age, 13% of them have hypertension, 6% have diabetes, and 5% have coronary artery diseases. He postulated that to obtain the best results in such patients, one should be aware of the presence of pre-existing cardiac diseases and aggressively treat them in order to prevent hypotension and hypoxemia [10]. The present study was conducted to investigate the connection between pre-existing risk factors/comorbidities and posttraumatic cardiac damage and/or the patients’ outcome.

## 2. Materials and Methods

This study was performed with 107 patients with multiple injuries, enrolled with an ISS ≥16 to the Level 1 Trauma Centre of the University Hospital of Frankfurt, Germany, and who were admitted from 2017–2020. Plasma samples were taken at admission in the emergency department, were immediately kept on ice and centrifuged by 3500× *g* for 15 min at 4 °C. The pre-existing cardiac risk factors, as well as pre-existing cardiac diseases, were recorded. Serum troponin T was determined by the routine laboratory using a high-sensitive electrochemiluminescence immunoassay (ECLIA, Roche, Rotkreuz, Switzerland). The patients’ collectives were subdivided into three groups according to the initial troponin T levels. The first group was defined by initial troponin T level ≤12 pg mL (“low troponin” group). The second group (“gray area” group) was defined by the troponin level between 13–49 pg/mL. The third group (“increased”) was defined by troponin level ≥ 50 pg/mL. Additionally, the routine blood diagnostic was performed to detect abnormalities that could represent a cardiac risk factor, for example, extreme high blood sugar levels (Accu Chek, Roche Diagnostics, Mannheim Germany), suggesting diabetes or high level of triglyceride (TAG, GPO PAP method, Roche Diagnostics, Mannheim, Germany) as an indicator of hypertriglyceridemia. All patients received an initial emergency diagnostic, including a chest CT-scan [11]. In three cases, a blunt cardiac contusion was detected via CT scan. A blunt chest trauma was defined by the occurrence of rib serial fractures, pneumothorax/tension pneumothorax, hematothorax/hematopneumothorax, sternum fracture, pericardial effusion or cardiac contusion. Human galectin-3, as well as human IL-33R, were detected in plasma samples using specific ELISA systems (human galectin-3 DuoSet ELISA, ST2/IL-33 DuoSet ELISA RD systems, Wiesbaden, Germany), following manufacture’s instruction. All clinical analyses were performed with ethical approval, given by the Independent Local Ethics Committee of the University of Frankfurt (approval ID 89/19).

All statistical analyses were conducted with Graph Pad-Prism 9, San Diego, CA, USA. The values are expressed as mean ± standard error of the mean (SEM). Normal distribution of the data was tested by a Kolmogoroff-Smirnow-Lilleforts-Test. Data were analyzed by a one-way ANOVA followed by Turkeys multiple comparison test. For the statistical analysis of two of the groups, *t*-test was applied. For the correlation analysis, linear correlation r was assessed by the Pearson test. Receiver operating characteristics (ROC) curve analysis was applied to estimate the area under the curve (AUC). Results were considered statistically significant when *p* ≤ 0.05.

## 3. Results

The study included 107 patients with an ISS ≥16. The patients were subdivided into three groups according to the troponin T concentration measured in the first blood samples taken on admission to the emergency room. In the “low troponin” group, 35 patients (Mean age 42 ± 15.2 years; ISS 25.7) were included. The “gray area” group also included 35 patients (Mean age 53 ± 20.9 years; ISS 30) and in the “increased” group, 37 patients (Mean age 48 ± 17.8 years; ISS 36.9) were included. Most of the patients were injured in traffic accidents (54.2%), the other injury causes include: falls from great heights (23.4%), crush injuries (4.7%), tripping (8.4%), train accidents (4.7%), gun shots/knife attacks (2.8%) and others (1.9%).

A cardiac contusion/pericardial effusion was observed in three patients in the initial CT scan. Chest trauma was a common injury in all three subgroups. In the “low troponin” group, 82.9% patients showed a chest trauma; in the “gray area” group, 65.7%; and in the “increased” group, 94.6%. A penetrating chest trauma was observed in two patients (Table 1).

The laboratory marker of cardiac damage, troponin T, is the current gold standard in the diagnostic of posttraumatic cardiac contusion. The group of patients with increased troponin level (≥50 pg/mL) showed significantly higher ISS and IL-6 concentration; spent a significantly longer time in the Intensive Care Unit (ICU)/Intermediate Care (IMC); and had a longer ventilation time, compared to patients without increased troponin T level (Figure 1A–D). Furthermore, troponin concentration was significantly higher in patients who needed catecholamines and in non-survivors (Figure 1E,F). We did not observe a correlation between troponin T and Interleukin (IL-) 6 concentration or the ISS (Figure 1G,H), and troponin was found to be a poor predictor of injury severity after trauma by means of ROC analysis (AUC 0.67) (data not shown).

In order to evaluate IL-33 R as a new marker of posttraumatic cardiac dysfunction, its concentration in the plasma of the included patients was measured. No difference in IL-33 R concentrations between the subgroups was detected (Figure 2A). Furthermore, no correlation between IL-33 R and troponin T concentrations was found (r^2^ = 0.097, data not shown). The IL-33 R concentration did not differ in survivors compared to non-survivors, but was significantly reduced in patients who needed catecholamines (Figure 2B,C). In addition, there was no correlation between the IL-33R level and ventilation time, time on ICU (data not shown), injury severity score (ISS) or IL-6 concentration (Figure 2E,F). IL-33 R was significantly increased in patients with cardiac comorbidities as compared to patients without (Figure 2D).

Next, galectin-3 was evaluated as a potential biomarker of cardiac damage in polytrauma patients. A significant increase in galectin-3 concentration was measured in the group of patients with increased troponin T concentration (≥50 pg/mL) (Figure 3A). However, no correlation between galectin-3 and troponin T concentrations was found (data not shown). Galectin-3 concentration did not differ significantly in survivors compared to non-survivors, as well as in patients who needed catecholamines compared to patients without this need (Figure 3B,C). There was also no correlation between galectin-3 plasma concentration and ventilation time or time on ICU/IMC (data not shown). A weak correlation between galectin-3 plasma concentration and injury severity score (ISS) (r^2^ = 0.31), as well as IL-6 concentration (r^2^ = 0.32), was found (Figure 3E,F). There is no difference between galectin-3 concentrations measured in patients with and without cardiac comorbidities (Figure 3D).

To analyze the potential link between pre-existing cardiac comorbidities or risk factors and post-traumatic damage to the heart, clinical and routine laboratory data were collected and investigated. As demonstrated in Table 3, in the ‘low troponin’ group, approximately 9% of patients had pre-existing cardiac comorbidity, while in the other groups, pre-existing cardiac disease was found in ~30% of patients. Hypertension was strongly (~20% of patients) represented in the “gray area” and “increased” troponin T patients’ groups. As demonstrated in Figure 4D, the systemic troponin T concentration was significantly increased in patients with pre-existing cardiac comorbidities in comparison to patients without (mean 59.77 pg/mL vs. mean 23.55 pg/mL, *p* < 0.05). Next to the cardiac comorbidities, other well-known risk factors of cardiac damage were analyzed in this study. Thus, diabetes was found to have been previously diagnosed in 3% of the “low troponin”, in 11% of “grey area” and in 5% of the “high troponin” group patients (Table 3). The blood sugar level measured at the point of admission in the emergency department was significantly increased in the patients from the “increased” troponin T-group. Another risk factor of cardiac comorbidities is hypercholesterolemia. Only one patient was previously diagnosed with hypercholesterolemia, while five patients took statins. In routine laboratory blood measurements in the emergency department, 17% of patients in both the “low troponin” and the “grey area” groups showed high triglycerides (TAGs) concentrations (≥200 mg/dL), while in the “increased” troponin T group, only 8% of patients had exceeded the norm values of TAGs. A correlation between TAGs or blood sugar levels and the troponin T levels at admission in the hospital was not found (Data not shown). The recorded Body Mass Index (BMI) did not differ between the three subgroups of patients (Figure 4C). No correlation between the risk factors and the troponin T level or the outcome was found (data not shown).

Significantly higher creatinine concentrations were measured in patients from the “increased” troponin T group compared to the “grey area” group patients (Mean values presented in Table 3), but a correlation between creatinine and troponin levels was not found in the present study.

As troponin T and IL-33R concentrations significantly differ in the patients with cardiac comorbidities, we analyzed this group of patients in detail. Figure 5A,B shows that BMI and initial blood sugar levels were significantly increased in these patients. Furthermore, in this subgroup, the troponin level was significantly increased in patients who needed support by catecholamines and non-survivors (Figure 5C,D). A moderate correlation was found between galectin-3 and ISS, and a weak correlation between galectin-3 and IL-6 concentration, as well as ventilation time, was found (Figure 5E,G). In addition, ventilation time in this group of patients was found to correlate weakly with troponin T concentration (Figure 5H).

## 4. Discussion

The aim of the present study was two-fold: to evaluate two new potential biomarkers (IL-33 R and galectin-3) of cardiac damage in polytrauma patients, and to assess pre-existing cardiac damage and/or risk factors as potential predictors of cardiac damage after polytrauma.

IL-33 R, a member of the interleukin 1 receptor family that is expressed in cardiomyocytes, is the receptor for interleukin-33, and is released by living cells in response to cell damage [12]. IL-33 R is suggested in the literature as a potential marker of cardiac damage in acute heart failure and acute coronary syndrome [9]. Furthermore, IL-33 R was described as a cardiac biomarker that is sensitive enough to distinguish between cardiovascular and non-cardiovascular causes of shortened breath [13]. A high baseline IL-33 value in patients with their first myocardial infarction was associated with progressive left ventricular volume indices dilatation and left ventricular ejection fraction deterioration [14]. Up to 24 h after admission to the hospital, IL-33R had increased in sepsis and trauma patients at the ICU [15]. In serum samples of polytrauma patients, the concentration of IL-33 R was elevated in comparison to patients with isolated chest trauma or healthy volunteers [16]. It is interesting to note that in the current study, patients with cardiac comorbidities had considerably higher levels of IL-33 R (Figure 2D), suggesting that this molecule may be used as a diagnostic tool to identify this patient subgroup. In the literature, pulmonary complications (ARDS and pneumonia) and mortality were associated with high systemic concentrations of IL-33 R [16]. Increased IL-33 R serum concentration in traumatic brain injury (TBI) patients significantly correlated with inflammation, severity, and prognosis, and IL-33 R was proposed as a novel marker for TBI [17]. In our study, non-survivors did not show increased level of IL-33 R concentrations (Figure 2B) and no further correlation of IL-33 R with the classical marker of traumatic inflammation IL-6 was found. Moreover, neither the troponin T concentration, nor one of the outcome parameters measured in the present study, were associated with an increase in IL-33 R. Overall, we found that IL-33 R is not associated with cardiac damage (detected by troponin measurements) or with any of the investigated outcome parameters, but is significantly increased in patients with cardiac comorbidities after polytrauma.

Galectin-3 is another new biomarker of early cardiac injury that has been proposed in the literature [8]. According to reports, it can be a helpful early biomarker of fibrosis, inflammation, and remodeling processes in cardiac injury, heart failure, and myocardial infarction [18]. Elevated levels of galectin-3 in patients with acute myocardial infarction were associated with the development of major adverse cardiovascular outcome [19]. Insulin resistance, left ventricular hypertrophy and circulating levels of galectin-3 were associated with a worsening of the diastolic function in morbidly obese patients. In experimental models, an increased level of galectin-3 was associated with acute respiratory distress syndrome (ARDS)-induced cardiac damage, [20] as well as acute myocardial infarction-induced cardiac remodeling [19]. The blockage of upregulated galectin-3 was found to decrease cardiovascular fibrosis and inflammation in a diet-induced obesity animal model [21]. In heart failure, systemic IL-33 R and galectin-3 were described as markers of left ventricular hypertrophy with reduced ejection fraction [22,23], as well as independent predictors for ischemia [24]. In the present study, galectin-3 showed no correlation with troponin T concentrations in polytrauma patients, but was significantly increased in the group of patients with troponin T levels ≥50 pg/mL compared to patients without a clinically relevant troponin T concentration. We found weak correlations between galectin-3 concentrations and the injury severity score, as well as IL-6 concentration. In the subgroup of patients with cardiac comorbidities, galectin-3 was not increased; interestingly, however, a moderate correlation among galectin-3 and the ISS, and a weak correlation with IL-6 concentration and ventilation time (Figure 5), were found. According to our findings, galectin-3 might be a possible marker of polytrauma severity, particularly in individuals with pre-existing comorbidities, however, it does not appear to be beneficial for detecting heart injury or pre-existing cardiac comorbidities. Based on our findings, galectin-3 seems not to be useful for detection of cardiac damage or pre-existing cardiac comorbidities, but could be suggested as a potential marker of polytrauma severity, especially in patients with pre-existing comorbidities.

The second part of our study examines the possible link between pre-existing comorbidities and/or cardiac risk factors (such as diabetes, hypertension, obesity or hypercholesterinaemia) and cardiac damage in polytrauma patients. Unfortunately, our list of risk factors was limited by the patient data we gathered. Significant risk factors, such as smoking and cardiovascular disorders in the patient’s family history, were not noted in the patient’s record, and were therefore not included in the study. We found that the initial blood sugar level was increased in the group of patients with a clinically relevant troponin T concentration. In the “increased” troponin T group, 34% of patients had initial blood sugar levels ≥200 mg/dL, although just 5.4% were formally diagnosed with diabetes. It has previously been shown that diabetic patients with a mild trauma (ISS 5–15) have an increased risk of hospital mortality, as well as serious infectious and cardiac complications. Diabetes was also associated with increased ventilation days, intensive care unit days and hospital stay length [25,26]. The higher prevalence of diabetes mellitus (50% vs. 10%) was found in patients who developed cardiac dysfunction after TBI [27]. In our study, no relation between initial blood sugar levels and survival was observed. Interestingly, compared to patients without this preload, patients with pre-existing cardiac comorbidities displayed considerably higher blood sugar levels (Figure 5). This finding is consistent with the information from textbooks that link high blood sugar and diabetes to the emergence of cardiac diseases.

Alongside the other factors, heart diseases and obesity were determined as independent risk factors of in-hospital mortality in trauma patients with an ISS >16 [28]. In a retrospective study in >200,000 trauma patients, obesity was found to correlate significantly with a higher risk of cardiac or respiratory complications (12% vs. 5.2%) [29]. Elderly and obese non-survivors of severe injury had a low cardiac index, resulting in reduced tissue oxygenation associated with organ failure and death [30]. However, other authors conclude that obesity does not appear to be a risk factor for adverse outcomes after blunt or penetrating trauma [31]. Our findings also did not find a link between BMI and/or the presence of obesity and the cardiac damage detected by troponin T, but they did demonstrate a relationship between pre-existing cardiac comorbidities and elevated BMI (Figure 5).

Another well-known risk factor of cardiovascular disease is hypercholesterolemia. In patients with severe trauma, the systemic analysis of hypercholesterolemia and its relationship with cardiac damage or outcome parameters have not yet been conducted. Decreased cholesterol precursor synthesis was seen in the first week following trauma, and it appeared to be the main factor contributing to hypocholesterolemia in patients with multiple trauma [32]. In ICU patients (after major surgery, multiple trauma, acute pancreatitis or septic shock), a low serum level of apolipoprotein A-I at admission was associated with an increase in SIRS criteria during the ICU stay [33]. Triacylglycerol, glycerol heads of phospholipids, and monosaturated fatty acids were found by Cohen et al. to be the most discriminating markers for identifying survivors from non-survivor’s trauma patients at submission time [34]. It is interesting to note that individuals with burn injuries had high triglycerides and low HDL, yet there was no change in their total estimated risk of cardiovascular disease [35]. We found that the TAGs level does not correlate with troponin T concentrations, and therefore does not appear to be a distinct risk factor for heart injury in polytrauma patients. To summarize the above discussed results, we did not find a relationship among pre-existing cardiac risk factors and cardiac damage after polytrauma in our collective of patients.

Next to the cardiac risk factors, we analyzed the relationship between cardiac comorbidities and cardiac damage in our polytrauma patients. The presence and number of premorbid conditions, anemia, age > 60, pulmonary diseases, cardiac or neurological diseases and ISS were previously determined to be death-predictors in patients with acute blunt chest trauma [36,37]. The cardiac comorbidities not only increase financial burden [38], but are a significant predictor of mortality in trauma patients [39,40]. Cardiovascular risk factors such as pre-injury warfarin usage, congestive heart failure, and pre-injury beta-blocker use, as well as combinations of these conditions, which result in the greatest mortality rates, increase the chance of trauma death [41,42]. The ‘metabolic syndrome’ (combination of unfavorable factors diabetes, hypertension and BMI >30 kg/m^2^) is strongly associated with increasing length of hospital stay, occurrence of cardiac arrest and myocardial infarction and is an independent death -predictor, which should be identified early in order to facilitate prompt multidisciplinary treatment [43]. Our findings that polytrauma patients with cardiac comorbidities have significantly higher concentrations of troponin T, IL-33R, higher BMI and initial sugar level are in accordance with this, and further emphasize the need for a specialized, multidisciplinary therapeutic strategy for patients with polytrauma and cardiac comorbidities.

## 5. Conclusions

Galectin-3 and IL-33 R, the two proteins under investigation in this study, were shown not to correlate with cardiac damage determined by troponin T concentration, making them unsuitable as cardiac damage-biomarkers in polytrauma patients. However, our data suggests that IL-33 R protein should be revised in future studies as a marker of cardiac comorbidities. We did not find a link between pre-existing cardiac risk factors and cardiac damage after polytrauma. In addition, our data indicate that patients with cardiac comorbidity represent a separate group of polytrauma patients characterized by higher concentrations of troponin T, IL-33 R, BMI and initial sugar level, and therefore need special attention and a differentiated, multidisciplinary treatment strategy.

## Figures and Tables

**Figure 1 jcm-11-06350-f001:**
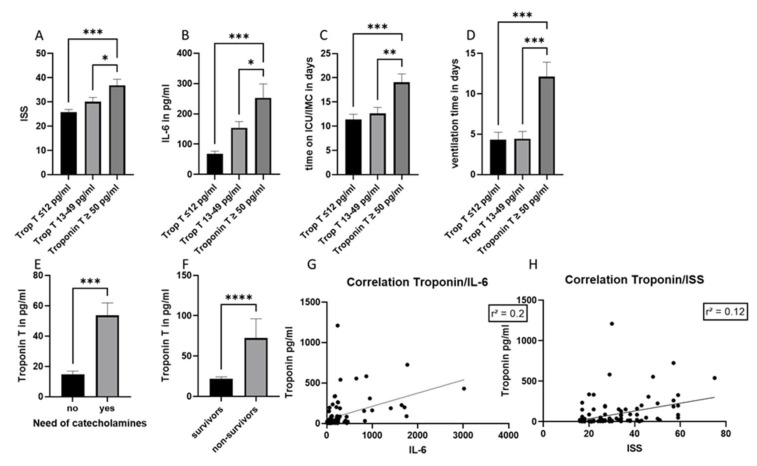
Relationship among ISS, IL-6, need in catecholamines and patient outcome with plasma troponin T. (**A**) Injury severity score (ISS) in the troponin subgroups of patients. (**B**) IL-6 concentration in the troponin subgroups. (**C**) Time on intensive care unit/intermediate care in days in the troponin subgroups. (**D**) Ventilation time in days in the troponin subgroups. (**E**) Troponin T concentrations in patients with (yes) and without (no) need in catecholamines. (**F**) Table 2 Troponin concentrations in survivors and in non-survivors. (**G**) Correlation analysis between troponin T and IL-6 measurements. (**H**) Correlation analysis between troponin T level and the ISS. *n* = 107 patients, * *p* ≤ 0.05, ** *p* ≤ 0.01; *** *p* ≤ 0.001; **** *p* ≤ 0.0001.

**Figure 2 jcm-11-06350-f002:**
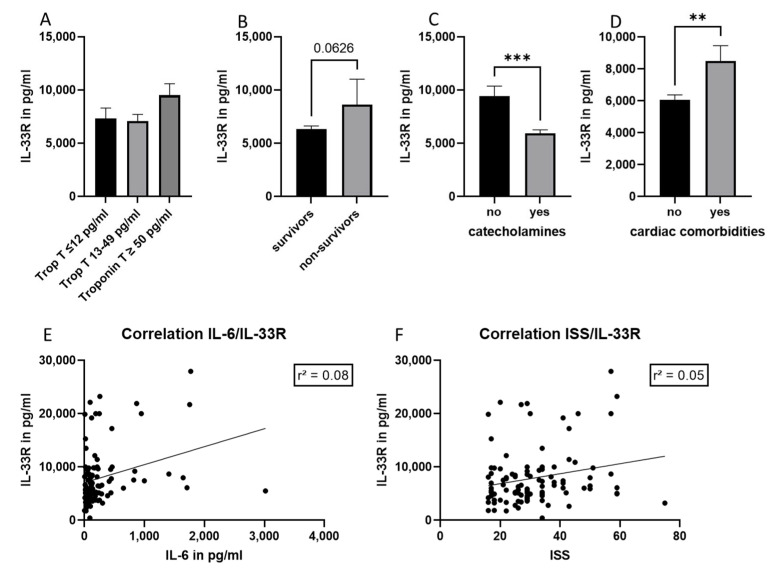
Interleukin 33 receptor (IL-33R) level in different groups of polytrauma patients. (**A**) IL-33R concentration in different troponin T patient subgroups. (**B**) IL-33R concentration in survivors and non-survivors. (**C**) IL-33R concentration in patients with (yes) and without (no) need in catecholamines. (**D**) IL-33R concentrations in patients with and without cardiac comorbidities. (**E**) Correlation analysis between IL-33R and IL-6 plasma concentrations. (**F**) Correlation analysis between IL-33R plasma concentrations and the injury severity score (ISS). *n* = 107 patients, ** *p* ≤ 0.01; *** *p* ≤ 0.001.

**Figure 3 jcm-11-06350-f003:**
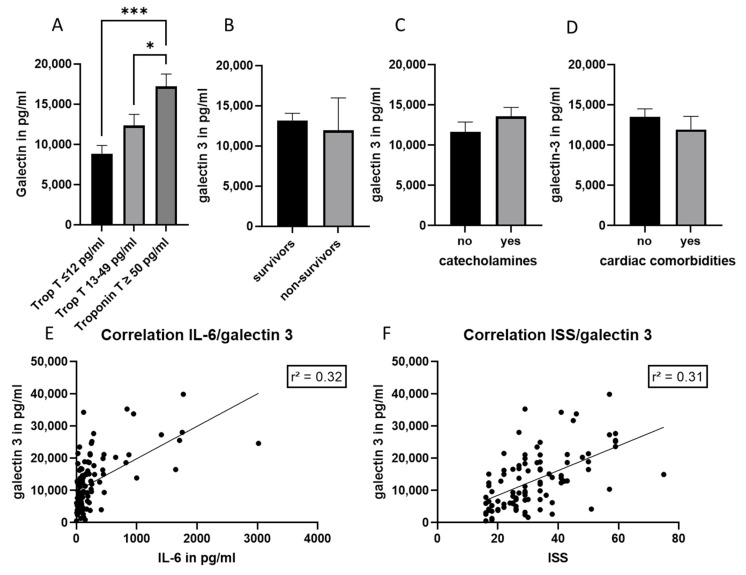
Galectin-3 concentration in different groups of polytrauma patients. (**A**) Galectin 3 concentration in the troponin T patients’ subgroups (≤12 pg/mL; between 13–49 pg/mL and ≥50 pg/mL). (**B**) Galectin-3 concentration in survivors and non-survivors. (**C**) Galectin-3 concentration in patients with (yes) and without (no) need in catecholamines. (**D**) Galectin-3 concentrations in patients with and without cardiac comorbidities. (**E**) Correlation analysis between galectin-3 and IL-6 plasma concentrations. (**F**) Correlation analysis between galectin-3 plasma concentration and the injury severity score (ISS). *n* = 107 patients, * *p* ≤ 0.05; *** *p* ≤ 0.001.

**Figure 4 jcm-11-06350-f004:**
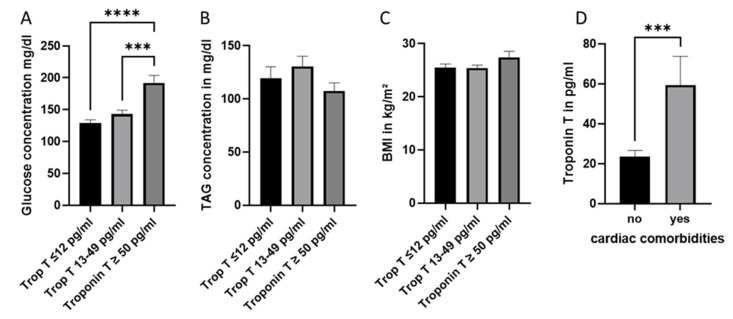
Cardiac risk factors and comorbidities in different groups of polytrauma patients. (**A**) Initial blood glucose concentration in the troponin subgroups (≤12 pg/mL; between 13–49 pg/mL and ≥50 pg/mL). (**B**) Initial triglyceride (TAG) blood concentrations in the troponin patients’ subgroups. (**C**) Body mass index (BMI) in the troponin subgroups. (**D**) Troponin T concentrations in patients with and without cardiac comorbidities. *n* = 107 patients, *** *p* ≤ 0.001 and **** *p* ≤ 0.0001.

**Figure 5 jcm-11-06350-f005:**
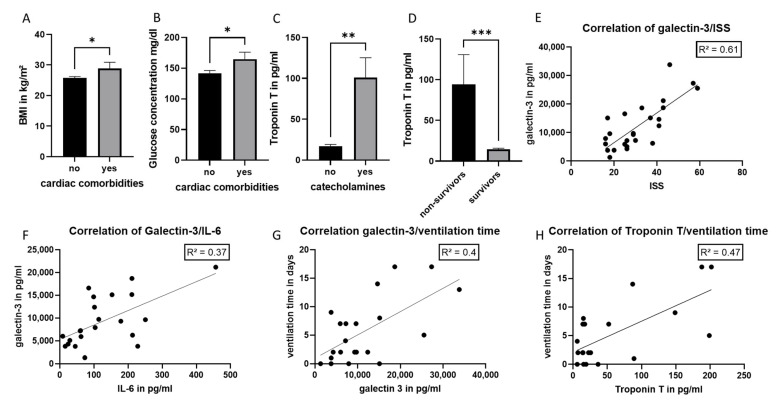
Patients with cardiac comorbidities–subgroup analysis. Body Mass Index (BMI) (**A**) and initial blood glucose concentration (**B**) in patients with (yes) and without (no) cardiac comorbidities. (**C**) Troponin T concentration in the subgroup of patients with cardiac comorbidities depending on the need of catecholamines. (**D**) Patients with cardiac comorbidities: Troponin T concentration in survivors and non-survivors. (**E**) Correlation analysis between galectin-3 and the Injury Severity Score (ISS). Correlation analysis between galectin-3 and IL-6 (**F**) or the ventilation time (**G**). (**H**) Correlation analysis between troponin T concentration and the ventilation time. *n* = 24 patients, * *p* ≤ 0.05; ** *p* ≤ 0.01; *** *p* ≤ 0.001.

**Table 1 jcm-11-06350-t001:** Cardiac damage evaluation in three groups of polytrauma patients.

Groups (Troponin T, pg/mL)	“Low” ≤12 pg/mL	“Gray Area” 13–49 pg/mL	“Increased” ≥50 pg/mL
Mean Troponin T (pg/mL)	6.32	23.83	205.1
Mean CK (U/L)	264.7	391.0	797.6
Mean CK-MB (U/L)	485.7	998.6	2467
Chest trauma	82.85%	65.71%	94.59%
Cardiac contusion/PE in CT scan	0%	0%	8.10% (*n* = 3)
Penetrating chest trauma	2.85% (*n* = 1)	0%	2.7% (*n* = 1)
Mean IL-6 (pg/mL)	67.63	153.6	252.6
Mean IL-10 (pg/mL)	58.43	117.9	222.5
Mean IL33R (pg/mL)	8862	12,360	17,208
Mean galectin-3 (pg/mL)	5896	6390	7853

**Table 2 jcm-11-06350-t002:** Polytrauma patients’ outcome, Troponin T subgroups.

Groups (Troponin T, pg/mL)	“Low” ≤12 pg/mL	“Gray Area” 13–49 pg/mL	“Increased” ≥50 pg/mL
Need for catecholamines	No 42.9% (*n* = 15)	No 42.8% (*n* = 15)	No 21.6% (*n* = 8)
	Yes 54.3% (*n* = 19)	Yes 48.6% (*n* = 17)	Yes 79.4% (*n* = 29)
	n.a. 2.8% (*n* = 1)	n.a. 8.6% (*n* = 3)	No 21.6% (*n* = 8)
Death	0%	8.57% (*n* = 3)	10.81% (*n* = 4)
Resuscitation	2.8% (*n* = 1)	5.71% (*n* = 2) (2 × päklinisch)	8.1% (*n* = 3) (3 × präklinisch)
Time on ICU/IMC in days	11.36	12.62	19.08
Ventilation time in days	4.3	4.4	12.12

**Table 3 jcm-11-06350-t003:** Cardiac comorbidities/risk factors in different groups of polytrauma patients.

Groups (Troponin T pg/mL)	“Low” ≤12 pg/mL	“Gray Area” 13–49 pg/mL	“Increased” ≥50 pg/mL
Age in years (mean)	42.09	53.54	48.19
Sex	F 17.14% (*n* = 6)	F 20% (*n* = 7)	F 17.14% (*n* = 6)
	M 8.86% (*n* = 29)	M 80% (*n* = 28)	M 8.86% (*n* = 29)
Cardiac comorbidities in general	8.57% (*n* = 3)	31.42% (*n* = 11)	27.02% (*n* = 10)
Hypertension	5.71% (*n* = 2)	20% (*n* = 7)	21.62% (*n* = 8)
Coronary heart disease	2.85% (*n* = 1)	0	5.40% (*n* = 2)
Arrhythmia	2.85% (*n* = 1)	11.42% (*n* = 4)	8.57% (*n* = 3)
Heart failure	0	8.57% (*n* = 3)	10.81 (*n* = 4)
Diagnosed with diabetes	2.85% (*n* = 1)	11.42% (*n* = 4)	5.40% (*n* = 2)
Blood glucose ≥ 200 mg/dL at admission	8.57% (*n* = 3)	2.85% (*n* = 1)	34.43% (*n* = 12)
Mean blood glucose levels (mg/dL)	129.1	142.8	191.8 *
Diagnosed with hypercholesterolemia	0%	0%	2.7% (*n* = 1)
Statin intake	5.71% (*n* = 2)	2.85% (*n* = 1)	5.4% (*n* = 2)
TAG ≥ 200 mg/dL	17.14% (*n* = 6)	17.14% (*n* = 6)	8.1% (*n* = 3)
Mean TAG (mg/dL)	119.5	130.4	107.5
Mean BMI (kg/m^2^)	25.43	25.42	27.39
Overweight (BMI ≥ 25 kg/m^2^)	20%	34.2%	13.5%
Adipositas Grad 1 (BMI ≥ 30 kg/m^2^)	5.71%	2.85%	13.5%
Adipositas Grad 2 (BMI ≥ 25 kg/m^2^)	2.85%	0%	2.7%
Adipositas Grad 3 (BMI ≥ 25 kg/m^2^)	2.85%	2.85%	2.7%
Diagnosed with Adipositas	2.85% (*n* = 1)	2.85% (*n* = 1)	5.4% (*n* = 2)
Mean creatinine level (mg/dL)	0.99	0.99	1.18

* *p* ≤ 0.05, significant difference compared to group “low”.

## Data Availability

Not applicable.

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
