# Peer review of "Evaluation of IL-33R and Galectin-3 as New Biomarkers of Cardiac Damage after Polytrauma—Association with Cardiac Comorbidities and Risk Factors"

_jcm, 2022, doi:10.3390/jcm11216350_

Round 1

Reviewer 1 Report

Dear authors,

I have several recommendations.

My suggestion is that you do not call the middle group of patients the “gray area”. The terminology “gray area” represents something inconclusive, that something is missing, and can not be easily defined. You have a group of patients with determined plasma concentration of TnT, and other markers, so it would be better for you to rename it appropriately. Did you measure hs-TNT?

Second, for a better understanding of your results, in lines e.g., 146,147, you state that “no correlation between galectin-3 and troponin T concentrations was found”, but in brackets, you still give the values, which I found unnecessary. However, in the following sentence (line 151) you state that “a weak correlation between galectin-3 plasma concentration and injury severity score (ISS), as well as IL-6 concentration, were found (Figure 3, E and F), but you don’t provide us the numbers. Please correct it, for better reading of your results.

Also, somewhere you write galectin, somewhere galectin-3 (e.g. figure 3), the same goes for the capital letters, please stay consistent.

Regarding the Table 3, please explain why did not provide a separate column with a "p" value? I still do not understand why do you mix commas (mean age) and points in the same Table (the gender?)

Some good insights, to improve the quality of your discussion, may be found in the papers very recently published, regarding the prognostic potential of sST2 and galectin-3 in cardiac pathology, such as:

Mitic VT, Stojanovic DR, Deljanin Ilic, et al. Cardiac Remodeling Biomarkers as Potential Circulating Markers of Left Ventricular Hypertrophy in Heart Failure with Preserved Ejection Fraction.Tohoku J Exp Med. 2020;250(4):233-242.

Stojanovic D, Mitic V, Stojanovic M, et al. The partnership between renalase and ejection fraction as a risk factor for increased cardiac remodeling biomarkers in chronic heart failure patients.  Curr Med Res Opin. 2020 ;36(6):909-919

Stojanovic D, Mitic V, Stojanovic M, et al.The Discriminatory Ability of Renalase and Biomarkers of Cardiac Remodeling for the Prediction of Ischemia in Chronic Heart Failure Patients With the Regard to the Ejection Fraction. Front Cardiovasc Med. 2021;8:691513.

Otherwise, the applied methodology is correct, and afterwards these corrections are made, the paper can be published.

Author Response

Response to Reviewer 1 comments:

Point 1: My suggestion is that you do not call the middle group of patients the “gray area”. The terminology “gray area” represents something inconclusive, that something is missing, and can not be easily defined. You have a group of patients with determined plasma concentration of TnT, and other markers, so it would be better for you to rename it appropriately.

Response: We agree with the reviewer that the term “gray area” might be confusing on the first sight. In the clinic, the troponin between 13-49pg/ml is officially defined as the “gray area”. If a patient shows a value in this “gray area” during routine laboratory diagnostic, the physician cannot disclose a cardiac damage and therefore has to determine the troponin concentration again after 6hrs (1). Because of the high clinical relevance, we decided to consider this “gray area” also in our patients collective.

(1) Weber B, Lackner I, Gebhard F, Miclau T, Kalbitz M: Trauma, a Matter of the Heart-Molecular Mechanism of Post-Traumatic Cardiac Dysfunction. Int J Mol Sci 22(2), 2021.

Point 2: Did you measure hs-TNT?

We measured the Troponin T concentration by using the ECLIA immunoassay of Roche, which is a high-sensitive method to detect Troponin T. To further highlight this fact, we changed the Material and Method section as follow: “Serum troponin T was determined by the routine laboratory using a high-sensitive electrochemiluminescence immunoassay (ECLIA, Roche, Rotkreuz, Switzerland).”

Point 3: Second, for a better understanding of your results, in lines e.g., 146,147, you state that “no correlation between galectin-3 and troponin T concentrations was found”, but in brackets, you still give the values, which I found unnecessary. However, in the following sentence (line 151) you state that “a weak correlation between galectin-3 plasma concentration and injury severity score (ISS), as well as IL-6 concentration, were found (Figure 3, E and F), but you don’t provide us the numbers. Please correct it, for better reading of your results.

Response: Thank you for this comment, we applied the necessary corrections.

Point 4: Also, somewhere you write galectin, somewhere galectin-3 (e.g. figure 3), the same goes for the capital letters, please stay consistent.

Response: We thank the reviewer for this comment. We checked the typing of “galectin-3” in the whole manuscript.

Point 5: Regarding the Table 3, please explain why did not provide a separate column with a "p" value?

Response: We did not provide a separate column with “p” values, as majority of the values provided in the table are percentage of patients in the group. We added “*” to that values, which differ significantly among the groups.  

Point 6: I still do not understand why do you mix commas (mean age) and points in the same Table (the gender?)

Response: Thank you for this helpful comment. We applied the necessary corrections in the Table 3.

Point 7: Some good insights, to improve the quality of your discussion, may be found in the papers very recently published, regarding the prognostic potential of sST2 and galectin-3 in cardiac pathology, such as:

Mitic VT, Stojanovic DR, Deljanin Ilic, et al. Cardiac Remodeling Biomarkers as Potential Circulating Markers of Left Ventricular Hypertrophy in Heart Failure with Preserved Ejection Fraction.Tohoku J Exp Med. 2020;250(4):233-242.

Stojanovic D, Mitic V, Stojanovic M, et al. The partnership between renalase and ejection fraction as a risk factor for increased cardiac remodeling biomarkers in chronic heart failure patients.  Curr Med Res Opin. 2020 ;36(6):909-919

Stojanovic D, Mitic V, Stojanovic M, et al.The Discriminatory Ability of Renalase and Biomarkers of Cardiac Remodeling for the Prediction of Ischemia in Chronic Heart Failure Patients With the Regard to the Ejection Fraction. Front Cardiovasc Med. 2021;8:691513.

Response: Thank you for the critical evaluation of our discussion section. We integrated the recommended literature in our discussion section as following:

“In heart failure, systemic IL-33R as well as the galectin-3 was described as marker of left ventricular hypertrophy with reduced ejection fraction [22,23] and further as independent predictors for ischemia [24].”

Reviewer 2 Report

The manuscript titled “Evaluation of IL-33R and Galectin-3 as new Biomarkers of Cardiac Damage after Polytrauma – Association with Cardiac Comorbidities and Risk Factors” by Weber et al. investigates IL-33R and galectin-3 as potential biomarkers of cardiac damage in polytrauma patients. Additionally, the authors evaluate if pre-existing cardiac comorbidities or risk factors can be used to predict cardiac damage after polytrauma. The paper provides a detailed analysis and examines correlations between IL-33R or galectin-3 levels with parameters of cardiac injury and conclude that the two proteins do not correlate with myocardial damage in polytrauma patients.

Major points:

1) The paper concludes that “IL-33R protein should be revised in future studies as a marker of cardiac comorbidities and galectin-3 protein as a marker of polytrauma severity in patients with pre-existing comorbidities”. As the authors acknowledge, the correlations between galectin-3 and trauma parameters are only moderate. The statement should be modified accordingly.

2) The second part of the paper would be stronger if a larger number of patients are included in the study, especially in the examination of the existing cardiac risk factors on post-traumatic myocardial damage.

Author Response

Response to Reviewer 1 comments:

Point 1: The paper concludes that “IL-33R protein should be revised in future studies as a marker of cardiac comorbidities and galectin-3 protein as a marker of polytrauma severity in patients with pre-existing comorbidities”. As the authors acknowledge, the correlations between galectin-3 and trauma parameters are only moderate. The statement should be modified accordingly.

Response: Thank you for this comment. We changed this sentence as following: “IL-33R protein should be revised in future studies as a marker of cardiac comorbidities.”

Point 2: The second part of the paper would be stronger if a larger number of patients are included in the study, especially in the examination of the existing cardiac risk factors on post-traumatic myocardial damage.

Response: Thank you for this comment. In the present study, we included all trauma patients admitted to our emergency room from 2017-2020, who has ISS ≥16 and had an initial measurement of troponin T in the routine laboratory diagnostic. We hope that future studies involving more patients will expand our findings.